# Thermal Post-Cross-Linking of Siloxane/Silsesquioxane Hybrids with Polycyclic Aromatic Units for Tailored Softening Behavior in High-Temperature Applications

**DOI:** 10.3390/molecules30173532

**Published:** 2025-08-29

**Authors:** Max Briesenick, Guido Kickelbick

**Affiliations:** Inorganic Solid-State Chemistry, Saarland University, Campus, Building C4 1, 66123 Saarbrücken, Germany; max.briesenick@uni-saarland.de

**Keywords:** silsesquioxanes, siloxanes, sol-gel, post-cross-linking, melting gels

## Abstract

Hybrid siloxane/silsesquioxane materials containing sterically demanding aromatic groups synthesized by hydrolysis and condensation suffer from incomplete cross-linking after thermal consolidation, limiting their thermal and mechanical performance. In this study, we systematically investigated a post-cross-linking strategy using various additives to enhance structural integrity and thermal stability. These include dimethyldimethoxysilane (DMDMS), diphenyldimethoxysilane (DPDMS) and phenyltrimethoxysilane (PTMS), as well as the organotin condensation catalyst di-*n*-butyltin diacetate (DBTA). Notably, we achieved thermal stability up to 453 °C and long-term transparency (up to 99%) at 200 °C with only little yellowing. Dynamic mechanical analysis demonstrated that post-cross-linking of precondensed siloxanes with PTMS, DPDMS, and DBTA enabled the formation of elastic materials exhibiting a rubbery plateau up to 200 °C. This behavior reflects enhanced structural rigidity and elasticity, which are essential for high-temperature applications. Our results show that high-temperature stability in siloxane/silsesquioxane materials is strongly influenced by factors such as the number of phenyl groups, cross-linking density, structural regularity, and degree of condensation. Most notably, the complete incorporation of a sterically demanding naphthyl-functionalized monomer during consolidation proved to be critical. Post-cross-linking significantly enhances all these parameters, which is essential for achieving robust thermal performance.

## 1. Introduction

Siloxanes are versatile materials employed across a broad spectrum of applications, ranging from architecture, healthcare, and aerospace to optical technologies such as organic light-emitting diodes (OLEDs), displays, and even biomedical devices like intraocular lenses [1,2,3,4,5]. Their widespread use is attributed to a combination of excellent properties, including flexibility, biocompatibility, gas permeability, high transparencies, thermal and UV stability, shock resistance, and good adhesion [1,4,6,7,8,9]. However, a significant limitation for optical applications such as LEDs is their relatively low refractive index (RI), typically between 1.4 and 1.5 [10]. Various strategies have been explored to enhance the RI, including the incorporation of polycyclic aromatic groups or titanium- or zirconium-containing compounds into the siloxane network [1,9,11,12,13]. While aromatic groups tend to increase viscosity due to enhanced friction, metal-based additives often lead to agglomeration and light scattering—undesirable effects in optical systems [14,15]. Siloxanes with diverse architectures can be synthesized through multiple approaches [4,16,17,18,19,20]. One such method involves the use of diphenylsilanediol and trialkoxysilanes to achieve high-RI materials [21,22,23]. By using acid-catalyzed hydrolysis and condensation reactions of aromatic di- and trialkoxysilanes, so-called melting gels can be produced [12,24,25]. After condensation, these materials can reversibly soften at temperatures around 110 °C. Once they have been treated at temperatures higher than 110 °C for a longer time, they consolidate. This means the cross-linking is complete and the melting gels can no longer be softened. The consolidation temperature is composition-dependent [26,27,28].

In earlier work, we synthesized siloxane/silsesquioxane hybrids via acid-catalyzed condensation of methyl-, phenyl-, and polycyclic aromatic-substituted alkoxysilanes. These materials exhibited high RIs, excellent thermal stability, and high optical transparency [12]. We also observed that the isomerism of naphthyl groups (1- vs. 2-naphthyl) influenced excimer formation, thermal stability, and glass transition temperature (T_g_), but had minimal impact on RI. Despite these advantages, prolonged thermal exposure led to yellowing and partial liquefaction, rendering the materials unsuitable for high-temperature applications. These include LED encapsulation, where temperatures may exceed 120 °C [29], protective coatings [30], electrical insulation for cables [31] and insulation casing for rocket cases [32]. This thermal instability appears to be specific to systems containing polycyclic aromatic substituents, as phenyl-based analogs consolidate into hard materials. However, even phenyl-rich systems can exhibit reduced cross-linking due to steric hindrance and a lower density of bridging oxygens, which increases the consolidation temperature [28,33]. These observations suggest that polycyclic aromatic groups hinder complete cross-linking, preventing full consolidation.

To address this limitation, we investigated post-cross-linking strategies using small reactive molecules to enhance network formation (Figure 1). Specifically, we employed methyl- and phenyl-substituted di- and trialkoxysilanes, which are common precursors in melting gel synthesis [26] and explored an alternative condensation route using alkyltin carboxylates such as di-*n*-butyltin diacetate (DBTA), known to vulcanize siloxanes under ambient conditions via Si–O–Sn bond formation [34,35]. These modifications allow for tuning of siloxane properties through variation in organic substituents, alkoxide substitution patterns, chain lengths, and molecular architecture, as demonstrated in previous studies on dimethylsilphenylene dimethylsiloxane oligomers [26,36,37,38,39]. In this work, we systematically investigate the effects of post-cross-linking on the thermal stability and cross-linking density of siloxane materials.

## 2. Results and Discussion

### 2.1. Synthetic Procedures

For the post-cross-linking investigations, we selected the previously studied siloxane system NaphMG, which incorporates 1-naphthyl substituents. This material was chosen due to its favorable combination of properties—namely, the lowest observed glass transition temperature (T_g_) among the tested systems, along with excellent processability and outstanding thermal stability following consolidation.

The NaphMG system is based on a 1:2:1 molar ratio of dimethoxyphenyl-(1-naphthyl)silane, dimethyldimethoxysilane, and phenyltrimethoxysilane (see Figure 1). The synthesis followed the protocol established in our previous work [12]. Briefly, the monomers were combined in a sealed headspace vial with methanol and aqueous hydrochloric acid (9 equivalents, pH = 1) and stirred at 45 °C for 72 h. After the initial hydrolysis and condensation, the reaction mixture was transferred to a beaker to allow gelation, followed by further condensation in a vacuum oven at 110 °C for 24 h. Subsequent thermal consolidation was performed at 200 °C for 72 h in a drying oven. The resulting material was solid at room temperature but retained the ability to soften upon heating, indicating incomplete cross-linking. The synthesis of the key monomer, dimethoxyphenyl-(1-naphthyl)silane, is described in detail in the Appendix A.

In contrast to phenyl-based silsesquioxane melting gels (PhMG), the NaphMG samples exhibited incomplete consolidation, as evidenced by a higher proportion of residual silanol and methoxy groups. This was confirmed through FTIR and NMR analysis, which revealed persistent hydroxyl and methoxy signals even after thermal treatment (Figure 2).

To improve the material properties, e.g., stiffness, T_g_, resistance to liquefaction, and thermal yellowing, a comparative study of various post-cross-linking strategies was conducted using the partially condensed NaphMG system (Figure 2). The goal was to enhance the degree of cross-linking and thereby stabilize the material under elevated temperatures.

The first approach involved post-cross-linking with dialkoxysilanes, which react with residual silanol groups via condensation. Two dialkoxysilanes were selected: dimethyldimethoxysilane (DMDMS) and diphenyldimethoxysilane (DPDMS). These compounds are commonly used in melting gel synthesis and offer more flexible cross-linking due to their bifunctional nature (Figure 2a,b) [26]. A second strategy employed phenyltrimethoxysilane (PTMS), a trialkoxysilane capable of forming a denser cross-linked network through its three reactive sites (Figure 2c). To isolate the effect of cross-linking density from other variables, a control sample with an equivalent initial cross-linking density was synthesized directly (Figure 2e), allowing for a direct comparison of thermal and mechanical properties.

Finally, we investigated the use of di-*n*-butyltin diacetate (DBTA), a known condensation catalyst that promotes siloxane network formation via Si–O–Sn linkages under ambient conditions (Figure 2d). This method offers a distinct mechanism for enhancing cross-linking and was evaluated alongside the alkoxysilane-based approaches. The following abbreviations were used to designate the synthesized samples: MG refers to Melting Gel, while Naph, Me_2_, Ph_2_, Ph, and Sn denote the use of dimethoxyphenyl-(1-naphthyl)silane, DMDMS, DPDMS, PTMS, and DBTA, respectively. The final part of each sample name indicates specific preparation details. For instance, NaphMG_5Ph_d refers to a sample prepared directly from monomers using 5 equivalents of PTMS, with “_d” indicating direct synthesis without post-cross-linking.

The partially condensed but unconsolidated NaphMG was still soluble and was therefore redissolved in methanol prior to post-cross-linking. Depending on the method, either alkoxysilane monomers and HCl (Figure 2a–c) or DBTA (Figure 2d) were added to initiate further condensation. The mixtures were stirred at elevated temperatures, allowed to gel, and subsequently subjected to thermal treatment to promote condensation (Figure 3). Due to the toxicity of the organotin compound, all pre-consolidation heat treatments were conducted in open headspace vials placed in a heat block within a fume hood, ensuring safe handling of the organotin compound and comparability with other samples Final consolidation was performed in a drying oven under ambient air (Figure 2a–c,e), except for the DBTA-containing sample, which was consolidated in a tube furnace under a wet argon atmosphere. The presence of moisture was essential, as polycondensation catalyzed by organotin compounds is ineffective under anhydrous conditions [34]. All resulting siloxane/silsesquioxane hybrids were solid at room temperature, with the exception of NaphMG_2Me_2_, which remained viscous.

Upon heating to 200 °C, all consolidated samples transitioned into a liquid state, apart from NaphMG_2Ph_2_ and NaphMG_4Ph, which exhibited softening while retaining elastic properties. To gain deeper insight into these observations, we conducted a series of analyses aimed at characterizing the thermal and mechanical behavior of the materials. Differential Scanning Calorimetry (DSC) and Dynamic Mechanical Analysis (DMA) were employed to determine the glass transition temperatures T_g_, elasticity, and softening behavior of the samples. In parallel, controlled heat treatment experiments at various temperatures were performed in a laboratory oven to assess thermal response under static conditions. To correlate the observed macroscopic behavior with structural differences, we carried out a comprehensive set of spectroscopic and structural analyses, including NMR, FTIR, and Powder X-ray Diffraction (PXRD). These techniques provided insights into the degree of cross-linking, residual functional groups, and potential ordering within the siloxane networks. Finally, the optical transparency and thermal stability of the consolidated siloxane materials were evaluated using UV–vis spectroscopy and Thermogravimetric Analysis (TGA), respectively. These measurements allowed us to assess the suitability of the materials for high-temperature optical applications.

### 2.2. Characterization of the Post-Cross-Linked Systems

#### 2.2.1. Differential Scanning Calorimetry

DSC was employed to determine the T_g_ of the consolidated siloxane samples. Measurements were conducted over a temperature range of 40 °C to 250 °C, except for NaphMG_2Me_2_, which was limited to 100 °C due to its expected low T_g_ (Table 1, Appendix A). Typically, siloxane systems containing only methyl and phenyl substituents consolidate within 24 h at temperatures up to 200 °C. However, due to the steric bulk of the naphthyl groups, which can hinder cross-linking, and to maintain consistency with previous studies, the consolidation time was extended to 72 h at 200 °C [12]. In fully consolidated systems, the T_g_ should be undetectable, reflecting a highly cross-linked, rigid network [26,28]. A comparison between the post-cross-linked samples and the initial NaphMG reveals that all post-treatment methods significantly influenced T_g_, either increasing or decreasing it depending on the cross-linking agent used. For example, replacing phenyl groups with methyl groups in the pristine samples led to a notable decrease in T_g_ from 68 °C (NaphMG_2Ph_2_) to −34 °C (NaphMG_2Me_2_), consistent with expectations based on the lower rigidity of methyl-substituted networks [40,41]. The use of the organotin catalyst DBTA also improved cross-linking, as evidenced by an increase in T_g_ from 5 °C (NaphMG) to 19 °C (NaphMG_Sn). Further comparison between NaphMG_4Ph and NaphMG_5Ph_d revealed T_g_ values of 33 °C and 67 °C, respectively, indicating substantial structural differences despite similar phenyl content. Notably, NaphMG_2Ph_2_ and NaphMG_5Ph_d exhibited the highest T_g_ values among all samples (68 °C and 67 °C), suggesting that the influence of bulky phenyl groups plays a dominant role in determining T_g_, even when cross-linking densities differ. Compared to other cured polysiloxanes containing methyl, phenyl, and carbazole groups, whose T_g_ values typically range from −83 °C to 56 °C [6,26,28], NaphMG_2Ph_2_ and NaphMG_5Ph_d demonstrate relatively high thermal rigidity. Since melting gels are expected to remain solid after consolidation, the presence of a detectable T_g_ indicates incomplete cross-linking [26,28,42].

In general, sterically demanding substituents on silicon atoms can hinder network formation, potentially requiring higher consolidation temperatures [26]. It is plausible that the exceptionally bulky naphthyl groups in the NaphMG system significantly obstruct complete consolidation, limiting the achievable cross-linking density.

Since all consolidated samples exhibited a detectable T_g_ in DSC measurements, it remained unclear whether these materials merely softened at elevated temperatures or underwent complete liquefaction. To address this, we conducted a preliminary thermal behavior study using stepwise heat treatment. Each consolidated sample was ground into a fine powder and subjected to incremental heating from room temperature to 200 °C in 25 °C steps, with each step maintained for 20 min (Appendix A). This approach allowed for visual assessment of softening and flow behavior under controlled conditions. The results revealed distinct thermal responses among the samples. NaphMG_Sn began to liquefy at approximately 100 °C, followed by NaphMG_5Ph_d, which showed signs of liquefaction at around 150 °C. In contrast, NaphMG_2Ph_2_ and NaphMG_4Ph remained solid throughout the entire temperature range, exhibiting only softening without flow, even at 200 °C. Interestingly, NaphMG_4Ph and NaphMG_5Ph_d, despite having identical compositions, displayed markedly different thermal behaviors. While NaphMG_4Ph had a significantly lower T_g_, it remained solid at elevated temperatures, unlike NaphMG_5Ph_d, which liquefied. This discrepancy highlights that T_g_ reflects the onset of increased polymer chain mobility, but not necessarily the transition to a liquid state. These findings underscore the critical role of post-cross-linking in determining the thermal response of the materials. Effective post-cross-linking can significantly enhance the structural integrity of the siloxane network, preventing liquefaction even at elevated temperatures. Accordingly, we conclude that NaphMG_2Ph_2_ and NaphMG_4Ph represent successfully cross-linked siloxane systems capable of maintaining their solid state up to at least 200 °C, making them promising candidates for high-temperature applications.

#### 2.2.2. Dynamic Mechanical Analysis (DMA)

To investigate the viscoelI confirmastic behavior of the consolidated siloxane samples, oscillatory rheometric measurements were performed using a plate–plate geometry. Each sample was placed between two plates, with the upper plate oscillating at a frequency of 1 Hz and an amplitude of 5%. The measurements were conducted over a temperature range starting from either 150 °C or 200 °C, depending on the sample’s viscosity, and cooled down to 35 °C (Figure 4, Appendix A). Due to the wide variation in viscosity among the samples, the starting temperatures were adjusted to prevent loss of contact between the plates at elevated temperatures. Additionally, initiating measurements at elevated temperatures was necessary to avoid fracturing the rigid samples at room temperature. This was particularly relevant for NaphMG_5Ph_d, which was too brittle to measure below 100 °C and was therefore only analyzed from 200 °C to 100 °C. The loss factor (tan δ) was calculated from the storage modulus (SM) and loss modulus (LM), and T_g_ was determined from the peak maximum of the tan δ curve (Figure 4c,d) [43,44]. For the base material NaphMG, T_g_ increased from 43 °C to 74 °C upon consolidation, and further to 80 °C with the addition of DBTA (NaphMG_Sn), indicating progressive network densification. However, the low viscosity of unconsolidated NaphMG at elevated temperatures led to repeated contact loss, resulting in artifacts in the data. Despite the increase in viscosity and SM upon consolidation (Figure 4a,b and Appendix A; Appendix A), the T_g_ value remained low. As expected, NaphMG_2Me_2_ exhibited the lowest viscosity and SM due to the flexibility imparted by methyl groups [45]. This sample could only be measured from 150 °C under cooling conditions, and no T_g_ was detected within the accessible range. Given its softness at room temperature, T_g_ lies below the measurement window as already indicated by DSC. In contrast, NaphMG_2Ph_2_ and NaphMG_4Ph exhibited T_g_ values of 124 °C and 67 °C, respectively, as determined by DMA, which are significantly higher than the corresponding DSC values (68 °C and 33 °C). The T_g_ values measured in DSC are lower than those measured in DMA, which has already been observed in the literature. The reason for this phenomenon is that the change in modulus against temperature is taken into account, which is a bulk phenomenon and is caused by the effect of the temperature on the flexibility of the monomer chains [46]. The overall T_g_ trend observed by DMA was: NaphMG_2Ph_2_ > NaphMG_Sn > NaphMG_cons. > NaphMG_4Ph > NaphMG_cond. This trend confirms that both consolidation and DBTA catalysis increase T_g_, while post-cross-linking with phenyl or methyl groups results in the highest and lowest T_g_ values, respectively. The unexpectedly low T_g_ of NaphMG_4Ph compared to NaphMG_5Ph_d, which both have the same composition, may be attributed to differences in network structure arising from post-cross-linking versus direct synthesis. It is important to note that these measurements were not performed using standard sample geometries (e.g., rods or thin films), but rather with tablet-shaped specimens compressed between oscillating plates, which may influence the absolute values obtained.

Comparing the viscosities at 200 °C (Appendix A and Appendix A) NaphMG_4Ph shows the highest values by far, followed by NaphMG_2Ph_2_, NaphMG_Sn and NaphMG_5Ph_d. The same trend was observed for the storage modulus (Figure 4a,b). NaphMG_4Ph reaches the highest rubbery plateau of all samples already at around 130 °C, followed by NaphMG_Sn reaching the third highest plateau at 160 °C and NaphMG_2Ph_2_ reaching the second highest plateau at around 190 °C, while NaphMG_5Ph_d does not reach a rubbery plateau at all, which shows that the former three keep their elasticity at high temperatures and form a cross-linked structure [43,47]. This was also confirmed during the thermal treatment experiment. Interestingly, NaphMG_Sn did turn liquid during this experiment either because it has the lowest elastic modulus of the three samples reaching a rubbery plateau or because the thermal treatment experiment was done with grinded samples instead of tablet-shaped and compressed samples as in the DMA.

In summary, the dynamic mechanical analysis showed an increase in viscosity and storage modulus by adding phenyl groups or increasing the cross-link density, which was to be expected since it is known that the modulus of elastomer increases with increasing cross-link density [48,49]. By comparing NaphMG_4Ph and NaphMG_5Ph_d it was also shown that post-cross-linking seems much more favorable to obtain siloxanes that are elastic and not liquid at high temperatures. Consequently, it was possible to prepare three samples, which show a rubbery plateau at 200 °C and therefore should keep their elasticity.

#### 2.2.3. Nuclear Magnetic Resonance (NMR) Spectroscopy

Since the consolidated samples behaved very differently in the previous measurements, we applied NMR spectroscopy to investigate potential correlations between the degree of condensation (DOC) and the observed T_g_s of the samples, as well as their liquefaction behavior. Therefore, we used ^29^Si- and ^13^C CP-MAS NMR spectroscopy and ^29^Si and ^13^C SP-MAS spectroscopy for samples NaphMG_2Ph_2_, NaphMG_4Ph and NaphMG_5Ph_d, as well as ^29^Si solution NMR measurements for samples NaphMG_cond., NaphMG_cons. and NaphMG_2Me_2_. NaphMG_Sn could neither be measured in liquid nor solid form, as it was not soluble in any solvent and due to its still slightly deformable consistency uniform rotation in the solid-state NMR was not possible. Furthermore, the DOC of all samples was determined (for further details, see Appendix A). A comparison of the results from ^29^Si CP-MAS and ^29^Si SP-MAS NMR spectroscopy (Table 2; Appendix A; Appendix A) reveals only minor differences in the signals for each monomer and for the DOC overall. Consequently, only the ^29^Si CP-MAS spectra were used for further analysis, as they offer superior resolution and allow for more precise fitting. Interestingly, the obtained DOC values (Table 2) indicate that any form of post-cross-linking leads to a reduction in DOC. This suggests that the introduction of additional aromatic groups and the increase in cross-link density may have a more significant impact on viscosity and softening behavior after consolidation. This is particularly relevant given that phenyl groups are known to increase monomeric friction, thereby raising viscosity [14]. Furthermore, this was also observed in the polyphenylsilsesquioxane melting gel (PhMG) from previous studies, which did not soften after consolidation, despite having a DOC of 83% [25]. A comparison between NaphMG_2Me_2_ and NaphMG_2Ph_2_ shows that the former exhibits a higher DOC, supporting our hypothesis that smaller substituents at the silicon atom promote more effective cross-linking. In the case of the monomer DMDMS, the increased DOC is likely attributed to a combination of steric and electronic effects [50,51,52]. Comparing the DOCs of NaphMG_4Ph and NaphMG_5Ph_d indicates that post-cross-linking leads to a higher DOC. The signal of 1-NaphPhSi(OMe)_2_ (D^2^, purple) for these two samples is particularly interesting (Figure 5) since it is almost 20% higher for NaphMG_4Ph. This indicates that post-cross-linking actually causes the remaining Si-OH and Si-OMe groups on the sterically hindered 1-NaphPhSi(OMe)_2_ to react further as we intended leading to an increasing number of bridging oxygens and therefore to a higher T_g_, which in turn has a significant impact on viscosity. This is particularly important in the case of the sterically very demanding 1-NaphPhSi(OMe)_2_, as phenyl groups are known to reduce the number of bridging oxygen atoms, which would otherwise weaken the siloxane network [28,33]. This observation was also made for NaphMG_2Ph_2_. Considering all observations, this shows that DOC is a factor, but not the only variable that determines the mechanical or thermal performance, such as high viscosity or liquefication of siloxanes. Further key factors include, for example, the density of the network, the steric hindrance or monomeric friction of the groups used. This is particularly noticeable for NaphMG_2Me_2_, which is soft at room temperature despite having the highest DOC, clearly showing the importance of the other characteristics. The PhMG also confirms this assumption, as it consolidates despite a significantly lower DOC of only 83%. Since only PTMS was used here, it can be assumed that the significantly higher cross-link density and high phenyl content are responsible for the desired properties. Therefore, it can be concluded that aromatic groups, cross-linking density and in our case the incorporation of sterically demanding groups into the siloxane network are also crucial. For the latter, we proved that post-cross-linking is much more effective, which is the key to preventing liquification upon elevated temperatures.

#### 2.2.4. Fourier Transform Infrared (FTIR) Spectroscopy

FTIR spectroscopy was performed on all samples as an additional characterization method to further study their condensation behavior, as well as the remaining hydroxy and methoxy groups (Figure 6 and Appendix A and Appendix A). Besides the aromatic groups and the methyl groups, the Si-O-Si stretching vibration is represented by two absorption bands, one from 1131 cm^−1^ to 996 cm^−1^ and the other one at 798 cm^−1^, indicating a successful network formation. This is further supported by the missing band of the methoxy groups at 1188 cm^−1^ [21,53,54]. During the condensation process, hydrolyzed species play an important role. Therefore the absorption bands at 3712 cm^−1^ to 3575 cm^−1^ (ν(OH_isolated_)), 3500 cm^−1^ to 3120 cm^−1^ (ν(OH_H-bonded_)) and 920 cm^−1^ to 890 cm^−1^ (ν(Si-OH)) are of great importance [53,55]. After the consolidation with additives, there are no hydrogen-bonded hydroxyl groups (3500-3120 cm^−1^) visible, indicating that the condensation is finished [12,25]. Comparing all samples, only NaphMG_2Me_2_ and NaphMG_Sn show no isolated OH groups anymore, leading to the assumption that these have the highest DOC. For NaphMG_2Me_2,_ this can be attributed to electronic and steric effects also mentioned in the last chapter [50,56]. NaphMG_Sn was further condensed with an organotin catalyst, which is known for room temperature vulcanizing (RTV) of siloxanes and therefore could explain the better condensation. In summary, the FTIR spectra show a successful condensation of all samples supporting the ^13^C CP-MAS NMR results.

The aforementioned Si-O-Si absorption band (1131–996 cm^−1^) can be used to gain more insight into the structures that could have formed inside the network (Figure 7). Since our samples are random networks formed of three to four different monomers, no clear statement can be made and only trends can be identified. For better comparison, all spectra were normalized using the Si-CH_3_ absorption band (1261 cm^−1^). The broad band ranging from 1115 cm^−1^ to 1037 cm^−1^ shows linear, branched, and cyclic structures and is not of much interest since they are random and cannot be further analyzed. The absorption bands at 1131 cm^−1^ and 1012 cm^−1^ on the other side can be attributed to ladder-like structures [57] where the band at higher wavenumbers becomes more intense when the regularity of the structure increases [58,59]. For our samples, this would mean that the amount of ladder-like structures and therefore the structural regularity is highest for NaphMG_2Ph_2_, NaphMG_5Ph_d and NaphMG_4Ph, which also contain the highest amount of phenyl groups. This shows that higher aryl content leads to more regular structures, which can arrange better in a solid sample, resulting in no liquification in the case of NaphMG_2Ph_2_ and NaphMG_4Ph at higher temperatures. However, since NaphMG_5Ph_d did turn liquid at higher temperatures, the high number of oxygen bridges from the better incorporated 1-NaphPhSi(OMe)_2_ shown in the NMR seems to be more important than the structural regularity.

#### 2.2.5. Powder X-Ray Diffraction (PXRD)

We applied PXRD measurements to obtain additional insight into the intra- and intermolecular spacing of our partly ladder-like structured siloxanes [17]. Two main peaks are identified in such polymers, one around 7° 2θ (d_1_), corresponding to the chain-to-chain distance, and a second one around 19° 2θ (d_2_) that is attributed to the intra-chain distance or the average thickness of the ladder. The ratio of their heights (R = I(d_1_/d_2_)) indicates the structural regularity of the ladder-like siloxane, which increases for higher d_1_ values [17,58,59]. All samples were measured in one piece without being pulverized (Figure 8, Appendix A). The first reflex increases from NaphMG_cond. to NaphMG_cons. to NaphMG_Sn. This shows an ongoing consolidation process, which leads to more Si-O-Si bonds by condensing free hydroxy and methoxy groups, thereby decreasing irregularities. For the former two, this was already seen via the DOC in the NMR spectra. The increasing DOC during the consolidation process leads to a denser and more rigid structure, resulting in an increase in d_1_ from 0.73 to 0.95 nm for these three samples. Masai et al. and Klein et al. for example also use a condensation method in which an acid is added first, followed by a base to complete hydrolysis and further cross-linking [26,60]. An increase in d_1_ from 0.80 to 1.04 nm, as well as in its intensity from NaphMG_2Me_2_ to NaphMG_2Ph_2_ indicates an increase in structural regularity and rigidity due to higher phenyl content, possibly due to π-π stacking. In addition to the IR spectra, where the increasing phenyl group content showed more ladder-like structures and therefore a higher structural regularity this time a clear trend can be seen. NaphMG_2Ph_2_ shows the highest regularity, followed by NaphMG_4Ph, while NaphMG_5Ph_d displays almost no regularity despite having the same number of aryl groups. It seems that structural regularity decreases if high amounts of PTMS are reacted at once as well as with increased cross-link density. The corresponding d_1_ positions of these three samples show the same trend indicating a denser, more rigid structure, which also gets supported by the NMR spectra showing an increasing DOC from NaphMG_5Ph_d to NaphMG_4Ph. The values of the d_2_ reflexes of all samples representing the intra-chain distance or thickness of the ladder are almost identical, which was to be expected, considering the ladders are always comprised of Si-O-Si bonds. In summary, the PXRD measurements show that the regularity of the siloxanes seems to increase with the amount of phenyl groups but decreases with cross-link density, at least when the same number of phenyl groups is present. Furthermore, adding PTMS to an existing siloxane seems to increase the regularity compared to adding all monomers at the start.

#### 2.2.6. Ultraviolet-Visible (UV-Vis) Spectroscopy

We studied the thermal stability and optical transparency of our siloxanes by measuring their transparency and yellowness index (YI) via UV-vis spectroscopy. The YI hereby describes the change from clear/white to a yellow color [61] and plays an important role regarding thermal stability, since high temperatures as well as UV light can lead to the creation of free radicals causing yellowing of the material [62]. All samples were doctor bladed onto glass slides after their condensation at a thickness of approximately 120 µm, heated in a vacuum drying oven for 24 h at 200 °C to remove any blistering, consolidated for 72 h at 200 °C and were kept for an additional 3 as well as 7 days at 200 °C in an oven (Figure 9, Appendix A). Due to the toxicity of the organotin catalyst, NaphMG_Sn was only measured after condensation, consolidation and after 7 d at 200 °C in a tube furnace (see Section 3). At 450 nm, all samples show transmittances of 98% or higher after condensation. After vacuum drying and consolidation, the transmittance for NaphMG_5Ph_d decreased the most, which can be attributed to the cracking of the sample due to its high PTMS content. In comparison, no cracks are visible for NaphMG_4Ph even though it has the same ratio of monomers. All other transmittances stayed the same or increased, which can be attributed to the shrinking of the films during consolidation. Comparing all completely heat-treated samples, apart from NaphMG_5Ph_d, those having higher phenyl ratios seem to lose less transmittance. This is consistent with the literature, according to which phenyl groups increase thermal stability in siloxanes [63,64,65]. If comparing the sample that was not further post-cross-linked (NaphMG) with the sample that was post-cross-linked using tin (NaphMG_Sn), it can be seen that the decrease in transmission of both samples after an additional 7 days at 200 °C is almost identical. The tin catalyst therefore does not appear to have a negative influence on thermal stability in terms of transmission. The YI measurements (Figure 9b) show a remarkably similar trend to the extent that samples with higher phenyl content show less yellowing, which was to be expected. Since all samples are transparent, loss in transmittance should result from yellowing. NaphMG_2Ph_2_, which already had the highest transparency, also shows the least yellowing. When comparing NaphMG_2Ph_2_ and NaphMG_4Ph with NaphMG_2Me_2_, NaphMG and NaphMG_Sn higher amounts of phenyl groups appear to reduce yellowing, which is consistent with higher thermal stabilities. A comparison of the samples NaphMG and NaphMG_Sn also shows an almost identical yellowness index, which again reveals that the tin catalyst does not have a negative influence on the thermal stability. In summary, the phenyl groups increase the thermal stability of siloxanes, resulting in higher transmittances and less yellowing after heat treatment. Additionally, post-cross-linking (NaphMG_4Ph) compared to reacting all monomers at once (NaphMG_5Ph_d) is beneficial not only to obtain siloxanes, which keep their elasticity at elevated temperatures but also in terms of transmittance and avoidance of cracks.

#### 2.2.7. Thermogravimetric Analysis

To further evaluate the thermal stability and decomposition behavior of our siloxanes TGA was performed. All samples were heated up to 800 °C under oxygen with a heating rate of 10 K min^−1^ (Figure 10, Table 3). NaphMG shows a T_95_ value of 374 °C. Further cross-linking using an organotin catalyst (NaphMG_Sn) leads to a slight decrease to 361 °C, while adding DMDMS (NaphMG_2Me_2_) does not change it. Using DPDMS (NaphMG_2Ph_2_) or PTMS (NaphMG_4Ph and NaphMG_5Ph_d), on the other hand, increases the T_95_ value to 419 °C, 390 °C and 453 °C, respectively. This shows that aromatic groups convey higher thermal stability [63] and that while Si-O-Si chains consisting of only D^2^ units are susceptible to thermal rearrangement degradation, while complex structures have better thermal stability [64].

All samples show two decomposition steps. The first is around 410 to 460 °C and the second between 540 and 570 °C. Only NaphMG_Sn shows a broad first decomposition step between 310 and 500 °C. These two decomposition steps were already observed for the samples in our previous work and further analyzed via TG-FTIR measurements [12]. These studies led to the conclusion that during the first decomposition step, aromatic groups are released due to Si-C bond cleavage. During the second decomposition step, small condensed species, possibly cyclic structures, were detected. Comparing the first decomposition step for NaphMG_4Ph and NaphMG_5Ph_d, the former shows much more Si-C bond cleavage, due to different structures that may have formed, which also explains the lower T_95_ value.

Residual masses have the same general trend as the T_95_ values. NaphMG_5Ph_d shows the highest value, followed by NaphMG_2Ph_2_, while NaphMG has by far the lowest residual mass with only 14%, which is due to the low amount of cross-linking [66,67]. Further cross-linking with the organotin catalyst (NaphMG_Sn) leads to a significant increase in the residual mass to 35%. Comparing NaphMG_2Me_2_ and NaphMG_2Ph_2_, the latter has a higher residual mass, although the sample has more carbon incorporated. One explanation could be the slightly lower degree of condensation, while another reason for this could be found in the structure. Looking at the two decomposition steps of these two samples, NaphMG_2Me_2_ lost most of its mass due to Si-C bond cleavage compared to NaphMG_2Ph_2_. It is known from literature that the insertion of phenyl groups increases the thermal stability of siloxanes. In addition, the thermo-oxidative stability of methyl group containing siloxanes is poor due to the oxidation of Si-CH_3_ at relatively low temperatures, which leads to the breaking of this bond [68,69,70,71,72]. Similar to the T_95_ values, NaphMG_4Ph also shows a lower residual mass than NaphMG_5Ph_d, which can also be explained by a significantly higher Si-C bond cleavage. In summary, the number of aromatic groups as well as the cross-linking density play a key role for the thermal stability of siloxanes. We also observed that the preparation route has a significant influence on their structures and thus on their decomposition behavior.

## 3. Experimental Section

### 3.1. Materials

Dialkoxysilane synthesis was carried out under inert atmosphere. Magnesium chips (99.9+%, Acros Organics, Geel, Belgium), 1-bromonaphthaline (97%, abcr GmbH, Karlsruhe, Germany) and phenyltrimethoxysilane (97%, abcr GmbH) were used without further purification. THF (99.8% HPLC grade, Fischer Chemical, Zurich, Switzerland) was purified in a MBraun SPS 5 solvent purification system (M. Braun Inertgas-Systeme GmbH, Garching, Germany). For polymer synthesis dimethoxydiphenylsilane (97%, Alfa Aesar, Ward Hill, MA, USA), dimethyldimethoxysilane (97%, abcr GmbH), phenyltrimethoxysilane (97%, abcr GmbH), di-n-butyltin diacetate (for synthesis, Merck-Schuchardt, Hohenbrunn, Germany) and MeOH (98%, BCD Chemie GmbH, Hamburg, Germany) were used without further purification. Hydrochloric acid (pH 1) was diluted from conc. HCl (for analysis, ACS BerndKraft, Duisburg, Germany) using demineralized water.

### 3.2. Instrumentation and Characterization Methods

Solid-state CP-MAS NMR spectra were recorded on an Avance III HD—Ascend 400WB spectrometer (Bruker Corporation, Billerica, MA, USA) using 4 mm inner diameter ZrO_2_ rotors with 13 kHz rotation frequency. The resonance frequencies were 100.67 MHz for ^13^C and 79.53 MHz for ^29^Si NMR spectra. Solid-state SP-MAS NMR spectra were recorded with the same instrumental setup.

NMR spectra in solution were recorded on an Avance III 300 MHz spectrometer (Bruker Corporation, Billerica, MA, USA) with 59.63 MHz for ^29^Si NMR spectra. The NMR samples were dissolved in chloroform-d (CDCl_3_) and 10^−2^ mol/l chromium(III)acetylacetonate as a relaxation agent was added. All spectra, except for the integrated solid-state ^29^Si CP-MAS and integrated solid-state ^29^Si SP MAS spectra, were plotted in MestReNova (v14.2.0-26256, Mestrelab Research, Santiago de Compostela, Spain) using the apodization function to adjust the signal-to-noise ratio. The ^29^Si solution NMR spectra were also adjusted using a multipoint baseline correction. The solid-state ^29^Si CP-MAS and ^29^Si SP MAS spectra were analyzed with OriginPro (Version 2021b. OriginLab Corporation, Northampton, MA, USA) and integrated using a Voigt function.

Fourier transform infrared (FTIR) spectra were recorded in attenuated total reflectance mode (ATR) on a Vertex 70 spectrometer (Bruker Optics, Ettlingen, Germany) from 4500 to 400 cm^−1^ for the consolidated siloxanes but only shown from 4000 to 400 cm^−1^ due to the absence of other signals at higher wavenumbers, each with a resolution of 4 cm^−1^ and 16 scans.

Powder X-ray diffraction (PXRD) patterns of the tablet-shaped samples were recorded at room temperature on a D8-A25-Advance diffractometer (Bruker AXS, Karlsruhe, Germany) in Bragg Brentano θ-θ geometry (goniometer radius 280 mm) with Cu K_α_ radiation (λ = 154.0596 pm). A 12 μm Ni foil working as a K_β_ filter and a variable divergence slit were mounted at the primary beam side. A Lynxeye detector with 192 channels and a variable slit diaphragm in front of it was used at the secondary beam side. Experiments were carried out in a 2θ range of 3−40° with a step size of 0.013° and a total scan time of 1 h. The recorded data was evaluated using TOPAS 5.0 (Bruker AXS, 2014, Karlsruhe, Germany) software, with the observed reflections being treated via single line fits and a background of 5 but no sample displacement.

For UV-vis transmittance measurements, all siloxanes were doctor-bladed onto glass slides (Microscope Slides, VWR, Radnor, PA, USA) at a thickness of approximately 120 µm, heated to 110 °C for 5 min in an oven to ensure a uniform film and measured after cooling. The measurement range was from 250 to 800 nm but is only presented from 300 to 800 nm due to the absorption of the class slides at low wavelengths. Consolidation of the siloxanes was performed in a vacuum drying oven at 200 °C for 24 h to prevent blistering and then for an additional 72 h at 200 °C in a drying oven. Further thermal treatment was carried out in a drying oven at 200 °C for an additional 7 days and additional measurements after 3 and 7 days. Consolidation of NaphMG_Sn was performed in a tube furnace under wet argon at 200 °C for 72 h, while thermal treatment was performed under synthetic air (N_2_/O_2_: 16/4) for an additional 7 d at 200 °C. All transmittance measurements were performed on a Lambda 750 instrument (Perkin Elmer Inc., Shelton, WA, USA) equipped with a 100 mm integration sphere with 2 nm increments and 0.2 s integration time. Yellowness index measurements were performed on the same samples at the same time intervals as the transmittance measurements from 380 to 780 nm, with 10 nm increments and 0.24 s integration time using the same instrumental setup.

Thermogravimetric measurements (TG) were carried out applying a TGA/DSC STARe System 1 (Mettler-Toledo, Schwerzenbach, Switzerland) applying a heating rate of 10 K min^−1^ between 25 and 800 °C using an oxygen gas flow of 40 mL min^−1^.

Differential scanning calorimetry was performed with a DSC 204 *F1 Phoenix* calorimeter (NETZSCH-Gerätebau GmbH, Selb, Germany) using aluminum crucibles with pierced lids under nitrogen/oxygen-flow (40/60 mL min^−1^), applying a heating rate of 10 K min^−1^. The temperature range was 100 to 250 °C, depending on the sample, and each sample was measured three times and the average calculated. The T_g_ was determined at the center point of the glass event.

Thermal treatment experiments were conducted with all solid samples by grinding them into powder and placing them on Teflon molds. They were heated in 25 °C steps from 50 to 200 °C for 20 min at each step in an oven.

Dynamic mechanical analysis (DMA) was performed using a MCR-301 rheometer with a CTD-450 convection heating system (Anton Paar GmbH, Graz, Austria) in oscillatory mode with a plate–plate geometry using a 25 mm PP25 measuring plate, an amplitude of 5%, a frequency of 1 Hz and a normal force value of 0. The samples were cooled from 150 °C or 200 °C to 35 °C with a cooling rate of 0.03 °C s^−1^, depending on the viscosity, to prevent them from losing contact with the upper plate.

### 3.3. Synthesis

#### 3.3.1. Synthesis of NaphMG and NaphMG_5Ph_d

The synthesis of NaphMG and NaphMG_5Ph_d was conducted applying a similar approach. The quantities of monomers, catalyst, and solvent are documented in Table 4. All monomers were weighed into a 50 mL headspace vial, dissolved in methanol and HCl (pH = 1) was added. After closing the headspace vial, the solution was stirred at 400 rpm at 45 °C for 72 h. The headspace vial was opened to gel overnight at room temperature and then heated to 110 °C for 24 h in a heat block.

#### 3.3.2. Synthesis of NaphMG_2Me_2_, NaphMG_2Ph_2_, NaphMG_4Ph and NaphMG_Sn

Syntheses of NaphMG_2Me_2_, NaphMG_2Ph_2_ and NaphMG_4Ph were carried out in a similar manner as already described above but with different amounts of solvent and HCl. The amount of added monomer, catalyst and solvent is documented in Table 5. In a 50 mL headspace vial, NaphMG (1 eq) was dissolved in methanol, the respective monomer and HCl were added, and the vial closed. After stirring at 400 rpm for 24 h at 45 °C, the headspace vial was opened to gel overnight at room temperature. After that the siloxanes were further condensed for 24 h at 110 °C in a heat block.

Synthesis of NaphMG_Sn was done in a slightly different way. In a 50 mL headspace vial, NaphMG (1 eq) was dissolved in methanol, di-*n*-butyltin diacetate (DBTA) was added and the vial closed. After stirring at 400 rpm for 24 h at 45 °C, the headspace vial was opened so that the reaction mixture could gel overnight at room temperature. After that, the sample was further condensed for 24 h at 110 °C in a heat block.

#### 3.3.3. Consolidation of the Siloxanes

All siloxanes except for NaphMG_Sn were consolidated for 72 h at 200 °C in a drying oven to obtain the cured siloxanes. Due to its toxicity, NaphMG_Sn was consolidated for 72 h at 200 °C in a tube furnace under wet argon.

## 4. Conclusions

In this study, various siloxanes were synthesized by post-cross-linking pre-condensed siloxane networks with dimethyldimethoxysilane, diphenyldimethoxysilane, phenyltrimethoxysilane, or di-*n*-butyltin diacetate. This approach yielded polymers with high degrees of condensation and enhanced thermal rigidity, effectively preventing liquefaction at elevated temperatures. The resulting materials incorporated naphthyl groups alongside varying proportions of phenyl and methyl substituents, leading to differences in cross-linking density. Among the synthesized samples, only NaphMG_2Ph_2_ and NaphMG_4Ph exhibited the desired thermal and mechanical properties. Comprehensive characterization using NMR, IR, and PXRD revealed that phenyl content, cross-linking density and structural regularity significantly influence the materials’ performance. Importantly, the post-cross-linking strategy proved superior to one-pot synthesis, which was especially seen when comparing the post-cross-linked sample (NaphMG_4Ph) with the direct approach (NaphMG_5Ph_d). Although both samples have the same composition, only the post-cross-linked sample showed increased elasticity and viscosity at elevated temperatures in the DMA measurements. An explanation for this is the higher structural regularity of NaphMG_4Ph, which was observed in PXRD measurements, as well as a higher DOC. In addition, the NMR measurements show an almost 20% improved incorporation of the sterically demanding 1-NaphPhSi(OMe)_2_. This contributes to a higher DOC and leads to a denser network. Furthermore, significantly more oxygen bridges are formed, which are essential for high viscosity and preventing liquefaction at high temperatures.

The consolidated siloxanes demonstrated high optical transparency (up to 99% after one week at 200 °C), minimal yellowing, T_95_ values up to 453 °C, and T_g_s ranging from −34 °C to 68 °C (DSC). Dynamic mechanical analysis further revealed T_g_ values up to 124 °C and the presence of rubbery plateaus at 200 °C in three samples, indicating retained elasticity. Thermal treatment confirmed that two of these materials remained solid and elastic at 200 °C, underscoring their potential for high-temperature applications.

Overall, this work demonstrates that post-cross-linking significantly influences the degree of condensation, structural integration, thermal stability, optical properties, and mechanical behavior of siloxanes. This strategy enables the design of materials with reduced brittleness at room temperature and sustained elasticity under thermal stress, making them promising candidates for advanced high-temperature applications.

## Data Availability

The original contributions presented in this study are included in the article/Appendix A. Further inquiries can be directed to the corresponding author(s).

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
