# Peer review of "Thermal Post-Cross-Linking of Siloxane/Silsesquioxane Hybrids with Polycyclic Aromatic Units for Tailored Softening Behavior in High-Temperature Applications"

_molecules, 2025, doi:10.3390/molecules30173532_

Round 1

Reviewer 1 Report

Comments and Suggestions for Authors

This article fills in some of the blanks in our knowledge of thermal stability in melting gels.  The family of gels that is studied includes the phenyl and diphenyl methoxysilanes plus the naphthyl substitutions.  The focus is on optical properties.  The goal is to find compositions that resist yellowing.  An added approach to completing the crosslinking is the use of the tin catalyst (DBTA).  The samples are thoroughly characterized and the crosslinking is quantified. 

The only concern I have is the use of the term liquefaction.  By definition, liquefaction is the process of becoming liquid.  However, the use of the term in practice is more often associated with the behavior of the ground under a building during an earthquake.  For that reason, as an English speaker, I find its use awkward.  Is it possible to use “softens” or “reduces viscosity”?  Liquification refers to a thermodynamic phase change, so I am not recommending its use.  In any case, it is not a major criticism, but it is something to think about.

Author Response

Comments 1: This article fills in some of the blanks in our knowledge of thermal stability in melting gels.  The family of gels that is studied includes the phenyl and diphenyl methoxysilanes plus the naphthyl substitutions.  The focus is on optical properties.  The goal is to find compositions that resist yellowing.  An added approach to completing the crosslinking is the use of the tin catalyst (DBTA).  The samples are thoroughly characterized, and the crosslinking is quantified. 

The only concern I have is the use of the term liquefaction.  By definition, liquefaction is the process of becoming liquid.  However, the use of the term in practice is more often associated with the behavior of the ground under a building during an earthquake.  For that reason, as an English speaker, I find its use awkward.  Is it possible to use “softens” or “reduces viscosity”?  Liquification refers to a thermodynamic phase change, so I am not recommending its use.  In any case, it is not a major criticism, but it is something to think about.

Response 1: Thank you very much for your suggestion. We appreciate your input and understand the intention behind the proposed wording. However, we chose the term “liquefy” to specifically describe the behavior of samples that were not successfully consolidated. While all samples exhibit softening and reduced viscosity at elevated temperatures, only the unconsolidated ones transition into a truly flowable or liquid-like state. Using a more general term would make it difficult to clearly distinguish between consolidated and unconsolidated materials in this context. Therefore, we believe “liquefy” most accurately reflects the observed behavior of these specific samples.

Reviewer 2 Report

Comments and Suggestions for Authors
  1. This manuscript reports the systematic exploration of thermal post-cross-linking strategies to enhance the thermal stability of siloxane/silsesquioxane hybrid materials containing polycyclic aromatic units. It effectively addresses the challenge of these materials softening or liquefying at high temperatures due to incomplete cross-linking, The results obtained through comprehensive synthesis and characterization can provide potential implications for high-temperature optical applications.

  1. This study uses a wide range of advanced analytical techniques, including DSC, DMA, NMR, FTIR, PXRD, UV-Vis, and TGA, to thoroughly investigate the changes in the materials' thermal, mechanical, structural, and optical properties. The research proves that the post-cross-linking strategy is significantly more effective than direct one-pot synthesis, especially for integrating sterically hindered alkoxysilanes like 1-NaphPhSi(OMe)2, which is key to preventing liquefaction at high temperatures.

  1. The abstract states that "only the two samples post cross-linked with phenyltrimethoxysilane and diphenyldimethoxysilane reached a rubbery plateau during heat treatment instead of becoming liquid". However, the DMA results section indicates that NaphMG_Sn also exhibited a rubbery plateau. While the paper attempts to explain this discrepancy by the difference in sample preparation for DMA vs. thermal treatment experiments, the statement in the abstract remains potentially misleading.
  2. The observation that post-cross-linking leads to a reduction in DOC for some samples is counter-intuitive for improving cross-linking and material rigidity. While the paper offers explanations (e.g., increased viscosity from aromatic groups, steric hindrance), a more in-depth discussion on the complex interplay of factors where DOC reduction might still lead to improved desired properties (e.g., preventing liquefaction) would be beneficial.
  3. It should be explained the mismatch between DSC and DMA Tg. For example, NaphMG_2Ph2 shows a Tg of 68 °C (DSC) vs. 124 °C (DMA)
  4. NaphMG_2Me2 has the highest degree of condensation (~93%) but is soft even at room temperature. Clarify that DOC is not the sole predictor of thermal/mechanical performance.
  5. Explain the inconsistency between NaphMG_4Ph and NaphMG_5Ph_d samples that have identical composition but behave differently in Tg and thermal flow tests.
  6. The DMA measurements were made at different temperatures for each sample, and DSC ranges vary. It would be better to standardize all the conditions.
  7. Unclear acronyms (e.g., MG, PhMG, NaphMG not immediately defined).

Author Response

Comments 1: This manuscript reports the systematic exploration of thermal post-cross-linking strategies to enhance the thermal stability of siloxane/silsesquioxane hybrid materials containing polycyclic aromatic units. It effectively addresses the challenge of these materials softening or liquefying at high temperatures due to incomplete cross-linking. The results obtained through comprehensive synthesis and characterization can provide potential implications for high-temperature optical applications.

This study uses a wide range of advanced analytical techniques, including DSC, DMA, NMR, FTIR, PXRD, UV-Vis, and TGA, to thoroughly investigate the changes in the materials' thermal, mechanical, structural, and optical properties. The research proves that the post-cross-linking strategy is significantly more effective than direct one-pot synthesis, especially for integrating sterically hindered alkoxysilanes like 1-NaphPhSi(OMe)2, which is key to preventing liquefaction at high temperatures.

The abstract states that "only the two samples post cross-linked with phenyltrimethoxysilane and diphenyldimethoxysilane reached a rubbery plateau during heat treatment instead of becoming liquid". However, the DMA results section indicates that NaphMG_Sn also exhibited a rubbery plateau. While the paper attempts to explain this discrepancy by the difference in sample preparation for DMA vs. thermal treatment experiments, the statement in the abstract remains potentially misleading.

Response 1: Thank you for your valuable comment. To ensure consistency with the findings from the DMA measurements, we have now included the third sample, NaphMG_Sn, in the statement as suggested.

Comments 2: The observation that post-cross-linking leads to a reduction in DOC for some samples is counter-intuitive for improving cross-linking and material rigidity. While the paper offers explanations (e.g., increased viscosity from aromatic groups, steric hindrance), a more in-depth discussion on the complex interplay of factors where DOC reduction might still lead to improved desired properties (e.g., preventing liquefaction) would be beneficial.

Response 2: We added a more detailed discussion to the NMR section.

Comments 3: It should be explained the mismatch between DSC and DMA Tg. For example, NaphMG_2Ph2 shows a Tg of 68 °C (DSC) vs. 124 °C (DMA)

Response 3: We added an explanation for the differences between DSC and DMA to the DMA section.

Comments 4: NaphMG_2Me2 has the highest degree of condensation (~93%) but is soft even at room temperature. Clarify that DOC is not the sole predictor of thermal/mechanical performance.

Response 4: We clarified that factors other than DOC also play an important role for thermal/mechanical performance in the NMR section.

Comments 5: Explain the inconsistency between NaphMG_4Ph and NaphMG_5Ph_d samples that have identical composition but behave differently in Tg and thermal flow tests.

Response 5: To clarify the findings, we have added an explanatory sentences in the NMR section as well as a paragraph in the conclusion that address the observations in more detail. We hope this addition improves the clarity and completeness of the discussion.

Comments 6: The DMA measurements were made at different temperatures for each sample, and DSC ranges vary. It would be better to standardize all the conditions.

Response 6: Due to the significantly varying mechanical properties of the samples—such as low viscosity at elevated temperatures and brittleness at lower temperatures—it was necessary to adjust the measurement conditions to ensure optimal data acquisition for each material. Nonetheless, we fully acknowledge the importance of standardized testing conditions and will aim to implement them as far as possible in future experiments to enhance comparability.

Comments 7: Unclear acronyms (e.g., MG, PhMG, NaphMG not immediately defined).

Response 7: We agree that introducing acronyms early on can improve clarity. However, we believe that presenting them alongside the corresponding structures enhances understanding and context for the reader. For this reason, we have opted to explain the acronyms at the point where the relevant structures are first shown.

Reviewer 3 Report

Comments and Suggestions for Authors

This work entitled “Thermal Post-Cross-linking of Siloxane/Silsesquioxane Hybrids with Polycyclic Aromatic Units for Tailored Softening Behavior in High-Temperature Applications” is somewhat interesting and has the potential for publication in this journal. It has some issues that could be addressed for possible acceptance. Here are my comments for revision:

  1. First of all, the abstract section is too long and the novelties of this work are not mentioned clearly. Therefore this section is suggested to be modified, in which the results findings should not included so much.
  2. The subscript throughout this work should be corrected written, including in the equations.
  3. There are some errors in the format of the references and he fonts in the Figures are too small, which should be at least as big as the caption of the figures.
  4. The English should also be polished since there have many written errors and grammar problems.
  5. Although the title is for high-temperature application, however, I did not find any potential for application, what is this work for should be highlight in the introduction as well. Please refer to the format of a research to concrete the introduction section, such as https://doi.org/10.1080/00268976.2025.2492391;
  6. The other sections are good and the findings are consistent with the analysis. I only wonder how do the authors control the temperature throughout the experiment, and how many is the errors and how to prevent it?

Author Response

Comments 1: This work entitled “Thermal Post-Cross-linking of Siloxane/Silsesquioxane Hybrids with Polycyclic Aromatic Units for Tailored Softening Behavior in High-Temperature Applications” is somewhat interesting and has the potential for publication in this journal. It has some issues that could be addressed for possible acceptance. Here are my comments for revision:

First of all, the abstract section is too long and the novelties of this work are not mentioned clearly. Therefore, this section is suggested to be modified, in which the results findings should not included so much.

Response 1: Thank you for your suggestion. In response, we have revised the abstract to more clearly highlight the novel aspects of our work, aiming to better communicate its significance and contribution to the field.

Comments 2: The subscript throughout this work should be corrected written, including in the equations.

Response 2: We corrected the subscript throughout the manuscript

Comments 3: There are some errors in the format of the references and he fonts in the Figures are too small, which should be at least as big as the caption of the figures.

Response 3: We have carefully reviewed and corrected the references as suggested. Additionally, we have updated the fonts in several figures to ensure consistency and improve readability.

Comments 4: The English should also be polished since there have many written errors and grammar problems.

Response 4: We have carefully reviewed the manuscript to correct typographical and grammatical errors.

Comments 5: Although the title is for high-temperature application, however, I did not find any potential for application, what is this work for should be highlight in the introduction as well. Please refer to the format of a research to concrete the introduction section, such as https://doi.org/10.1080/00268976.2025.2492391;

Response 5: Thank you for your suggestion. In response, we have expanded the introduction by adding additional examples of high-temperature applications of silicones to better contextualize the relevance and potential impact of our study.

Comments 6: The other sections are good and the findings are consistent with the analysis. I only wonder how do the authors control the temperature throughout the experiment, and how many is the errors and how to prevent it?

Response 6: The synthesis temperatures were carefully controlled using a heating stirrer in combination with a custom-designed aluminum attachment for headspace vials. Subsequent steps - including condensation, consolidation, and aging - were carried out in a drying oven. All temperature-dependent measurements (DMA, DSC, and TGA) were performed using appropriately calibrated instruments to ensure accuracy and reproducibility.

Round 2

Reviewer 3 Report

Comments and Suggestions for Authors

This version can be accepted.